# Lighting in the Home and Health: A Systematic Review

**DOI:** 10.3390/ijerph18020609

**Published:** 2021-01-12

**Authors:** Oluwapelumi Osibona, Bethlehem D. Solomon, Daniela Fecht

**Affiliations:** MRC Centre for Environment and Health, School of Public Health, Imperial College London, London W2 1PG, UK; oluwapelumi.osibona16@imperial.ac.uk (O.O.); b.solomon@imperial.ac.uk (B.D.S.)

**Keywords:** light, illumination, housing, residential, home, health

## Abstract

Poor housing is an important determinant of poor health. One key aspect of housing quality is lighting. Light is important for visual performance and safety, and also plays a vital role in regulating human physiological functions. This review aims to synthesise existing evidence on the relationship between lighting in the home and health and recommends areas for future research. Three databases were searched for relevant literature using pre-defined inclusion criteria. Study quality was assessed using the Newcastle Ottawa Scale. Extracted data were qualitatively synthesised according to type of lighting (natural light, artificial light and light at night) and stratified by broad health domains (physical, mental and sleep health). Of the 4043 records retrieved, 28 studies met the inclusion criteria. There was considerable heterogeneity in light exposure metrics used and specific health outcome assessed by the studies. Lighting in the home can negatively affect health but the current evidence base is limited to a small number of studies in different domains of light and health. Further research surrounding specific health outcomes is required to better inform housing quality assessments and lighting practises in the home.

## 1. Introduction

The right to adequate housing is a recognized international human right [1]. The World Health Organisation (WHO) defines healthy housing as one that encourages a state of complete physical, mental and social well-being [2]. People living in inadequate housing are at greater risk of ill health [3,4,5] and inadequate housing conditions are one of the main drivers of health inequalities [2]. Adequate housing is commonly assessed based on housing quality [4,5] which encompasses a wide variety of factors including: crowding and home safety, mould and dampness; temperature and humidity; ventilation and insulation; sanitation; indoor air and noise pollution; radon, asbestos and lead exposure; and lighting [2,4]. Many housing quality factors co-exist within the home, placing occupants at a greater risk of multiple health problems. Housing quality is associated with different health outcomes including developmental, chronic and acute conditions [5,6]. Many housing quality factors are widely studied, for example, mould and dampness [7]; crowding [8]; and lead exposure [5]. Others, however, are understudied despite their potential to impact health. One of the less studied housing quality factors is lighting in the home.

Adequate lighting is needed for visual performance and safety, and to reduce falls and injuries. Light is also highly essential for health and well-being [9,10,11] through the regulation of bodily functions [12]. Light plays an important role in the function of the nervous and endocrine systems and the secretion of hormones such as melatonin. Melatonin is released by the pineal gland in a 24-h cycle according to how much light is received, regulating the body’s circadian rhythm. In regular sleep-wake cycles, the hormone is highest at night in the dark promoting healthy sleep and lowest during daylight promoting alertness. Disruption to these rhythms caused by a lack of daylight exposure during the day and exposure to bright lights during the night constitutes as improper light exposure which affects health [9,13].

The importance of light on health is further demonstrated through its therapeutic effects. Symptoms of seasonal affective disorder and other types of depression have been shown to be effectively reduced by both natural and artificial light therapy [14,15,16,17]. Before the discovery of antibiotics, sunlight played a significant role in infection control and preventing the spread of disease in buildings [18,19]. Even today, forms of artificial light are effectively being used in hospital settings to reduce infection transmission [20,21].

Lighting within the home encompasses different types of light. For instance, homes may be illuminated by natural light through windows and supplemented with artificial light sources during the day, with artificial lighting continuing into the night. As such, there is a need to understand the impact of the various types of lighting in the home on the health of residents. A limited number of systematic reviews have previously explored the impact of lighting on the elderly [22] and the effects of sunlight [23] and light at night [24] on health in certain settings such as care homes. A systematic review that synthesises the evidence of health impacts from different types of lighting in the home is lacking.

This study aims to systematically review the literature and synthesise the existing evidence on associations of lighting in the home from natural light, artificial light and light at night with a broad range of health outcomes. Light is defined in its broadest sense to gather evidence on different aspects of lighting in the home and its effect on health, including lack of light, different types of illuminance and light as potential hazard source, for example, from indoor air pollution and burns. This allows identifying areas for further research and implications for policy development.

## 2. Materials and Methods

### 2.1. Protocol

This systematic review on lighting in the home and health was carried out according to the Preferred Reporting Items for Systematic Reviews and Meta-Analyses (PRISMA) reporting guidelines.

### 2.2. Search Strategy

Three databases (Embase, MEDLINE and Scopus) were searched for published literature from inception till February 2020 with no language restriction. The search strategy combined search terms for light exposure, health outcome and study setting. Keywords were identified following a preliminary screening of relevant literature in MEDLINE and relate to sunlight, nightlight, daylight, natural light, artificial light, illumination, residential light, domestic light, health, mental health, and falls. The full search strategy is outlined in Appendix A.

### 2.3. Eligibility Criteria

#### 2.3.1. Study Type 

Observational and intervention studies were included. Case reports and systematic reviews were excluded.

#### 2.3.2. Population

No restrictions were applied to population characteristics. Studies carried out in a home setting (e.g., domestic, student and nursing homes) were included, those in institutional (e.g., hospital, prison) and experimental settings were excluded. Studies with mixed settings (e.g., nursing homes and hospitals) that do not differentiate the results between the two settings were excluded.

#### 2.3.3. Exposure

Any lighting exposure within the home, including sunlight, artificial light and light at night using subjective or objective exposure metrics (e.g., lighting perception versus measurements) were included. Lighting outside of the home (e.g., streetlights) as well as lighting from electronic devices (e.g., televisions and mobile phone devices) were excluded. Intervention studies that prescribe light treatment for certain time periods were excluded as these reflect therapeutic lighting which is not the exposure of interest.

#### 2.3.4. Outcome Measures

Studies including at least one health outcome in relation to the exposure, either self-reported or measured were included. Studies reporting on melatonin, used as a biomarker for circadian dysregulation [25], were also included.

#### 2.3.5. Publication Type

Peer-reviewed studies published in academic journals were included. Conference abstracts, editorials and letters as well as studies not available in the English language were excluded.

### 2.4. Study Selection

Title and abstracts were screened for eligibility based on the inclusion criteria by the first author (OO), with a random sample of 16% independently screened by the second author (BS). Acceptable concordance was predefined as agreement on at least 90% of ratings, a concordance of 95% was achieved. Full text was independently screened by OO and BS based on the eligibility criteria. At both stages, any discrepancies were resolved by discussion and consensus with the last author (DF).

### 2.5. Data Extraction

Information was extracted from eligible studies relating to study characteristics (including author, year, study design, and sample size), participant characteristics, lighting exposure (including type, assessment and measurement) and the health outcome (including data collection method and measure of association) using a customised data collection form.

### 2.6. Study Quality Assessment

Studies were evaluated using the Newcastle-Ottawa Scale for assessing the quality of non-randomised studies (e.g., cohort and case-control) [26]. The scale was adapted for assessment of intervention studies, as the success of the intervention was not the interest of this review but the association with the outcome at the different lighting environments, and cross-sectional studies. The scale awards stars in three categories: selection of the groups of study (maximum four stars), comparability (maximum two stars) and assessment of the outcome (maximum three stars), for a maximum score of nine. An overall score of 0–3 was defined as poor quality, 4–6 as fair quality, and 7–9 as good quality.

### 2.7. Data Analysis

There was considerable heterogeneity in the types of lighting, the exposure metrics used to quantify light and health outcomes across studies, thus it was not feasible to summarise findings via meta-analysis. Instead, studies were qualitatively synthesised, stratifying by type of lighting exposure and health outcomes. Type of lighting was categorised into: (i) natural light: light produced naturally by the sun; (ii) artificial light: fuel-based (e.g., kerosene lamps) and electric lighting; and (iii) light at night: lighting occurring specifically during the evening and night-time period within the home.

Specific health outcomes were categorised into broad health domains: physical, mental and sleep health. In studies where the reported health outcome was overall health or personal health, these were reported under general health.

## 3. Results

The PRISMA flow diagram (Figure 1) outlines the process of the literature search on lighting in the home and health and consequent screening. The search initially yielded 4043 potentially relevant studies, of which 3120 were unique. 2866 studies that did not meet the inclusion criteria were excluded during title and abstract screening. The full text of the remaining 254 studies were sought for further assessment, 226 studies were excluded, most commonly due to incorrect publication type such as posters and abstracts. Twenty-eight studies were eligible for inclusion in the review based on the pre-defined criteria.

### 3.1. Study Characteristics

Key characteristics of studies included in the review are outlined in Table 1. Of the included studies, 20 focused on high income countries, including 11 studies from Japan; eight studies were carried out in Low-Middle Income Countries (LMIC). Combined, studies included a total of 965,056 participants, with sample sizes ranging from 17 to 932,341 [27] participants. Included studies varied substantially in study design ranging from cross-sectional studies (*n* = 14) case-control studies (*n* = 8), to cross-over studies (*n* = 2), longitudinal studies (*n* = 2) and randomised trials (*n* = 2).

Five studies observed metrics related to natural light (including sunlight), twelve at artificial light (including types of electrical light and fuel-based lighting) and ten at light at night (including intensity and exposure). One study assessed health outcomes in relation to artificial light and light at night and, as such, was included in analysis for both categories. There was great heterogeneity in the exposure metrics used to quantify lighting in the home across the studies. Twenty-three studies measured the amount of light using either objective measurements (*n* = 12), for example, using a light meter, or subjective measurements reported by study participants (*n* = 11), for example, using questionnaires or interviews.

A broad range of health outcomes were analysed by the included studies (Table 2). Fifteen studies reported on physical health, two on mental health, three on sleep health and two studies on general health. Six studies reported on two or more health domains. Health outcomes were either self-reported (*n* = 11), assessed by clinical examination (*n* = 7), from health record linkage (*n* = 3) or measured (actigraphy) (*n* = 1). One study used both clinical examination and clinical investigation and five studies used mixed methods (self-reported and diagnostic).

### 3.2. Methodological Quality Assessment

The quality score of the studies is shown in Table 1 and the full assessment details are presented in Appendix A. Overall, ten studies were deemed to be of good quality and eighteen of fair quality. No studies were rated as poor quality. Reasons for fair quality mainly related to the lighting exposure assessments which use unvalidated, subjective measurements. Furthermore, studies frequently lacked an independent assessment of health outcomes, instead using self-reported health, which increases the risk of bias affecting study quality.

### 3.3. Summary of Findings

Table 1 summarises the main findings for all studies stratified by the three lighting in the home categories natural light, artificial light, and light at night.

#### 3.3.1. Natural Light

There is strong evidence that natural light has a positive impact on health. All five studies exploring relationships of natural light with health outcomes found an effect on at least one of the health outcomes investigated.

##### Physical Health 

Two studies [28,29] explored the association of natural light exposure with infectious diseases. Rahayu et al. found individuals with household sunlight exposure were 94% less likely (Odds Ratio (OR) = 0.06, 95% confidence interval (CI) 0.00–0.67) to be diagnosed with tuberculosis than those without [28]. Kumar et al. observed an increased risk of leprosy associated with insufficient natural light exposure (OR = 1.57, 95% CI 0.84–2.88) [29]. The presence of natural light also associated with injuries. In a study on natural light and fall incidents, individuals reporting falls were more likely to also report inadequate natural lighting in the home (OR = 1.50, 95% CI 1.2–1.9) [30]. One study among 76–90-year-old elderly home-dwellers, however, measured daytime illuminance at two points within the home (the living room and bedroom) and reported no evidence of a relationship with physical health [31].

##### Mental Health

A cross-sectional study in eight European cities with 6017 participants assessed the effect of inadequate home lighting on depression [30]. Self-reported inadequate natural light increased the likelihood of reporting depression (OR = 1.40, 95% CI 1.2–1.7) [30]. Two studies reported a positive association between illumination and reduced self-reported depression scores using different exposure metrics: one study utilised time spent in high levels of daytime illumination (≥400 lux) in Japanese elders aged 76–90 years [31] and the other study used measurements of morning illumination during first 4-h period after arising in post-menopausal women in USA [32]. Reduced window covering in the morning was also associated with greater light influx and better depression scores in the latter study [32].

##### Sleep Health

Youngstedt et al. explored the relationship between morning illumination and sleep. The study assessed sleep using sleep duration (a measure of the total hours of sleep), sleep quality (a measure of how well one sleeps) and sleep latency (how long it takes to fall asleep). They found improved sleep (encompassing sleep duration, sleep quality and sleep latency) associated with morning illumination [32].

#### 3.3.2. Artificial Light

Health outcomes were explored in association with different sources of artificial lighting in five studies, different electrical lighting equipment in two studies, the adequacy of the lighting present in three studies and adjustments to existing electrical lighting in three intervention studies. There is strong evidence that certain types of artificial light have a positive impact on health. Of the 13 studies exploring relationships with health outcomes, only one study found no effect with at least one health outcome.

##### General Health

A study in Uganda found homes using “d.light D20g”, a type of home solar lighting system, had higher scores for personal health (35.2 percentage points increase) compared to controls using low-quality light sources, mainly kerosene candles [33]. Two intervention studies explored adjustments to living room lighting using lamps to increase the intensity of light present. A study in Swedish adults with low vision found the provision of a singular floor lamp in the individual’s living room increased quality of life (including self-reported general health and improved depressed mood ratings) [34]. A study amongst home-dwelling elderly evaluated the effect of supplying lamps to improve their living room lighting to 200 lux. The study found no effects on visual problems on general health [35].

##### Physical Health

Three studies compared use of kerosene-based lighting with electric lighting on respiratory infections. One study found no association between household light source and tuberculosis [36]. Two studies [27,37], both in child populations in India, found an association between acute respiratory infections and kerosene fuel lighting. Savitha et al. reported that households with diagnosed cases of acute lower respiratory infection had a higher prevalence of kerosene lamp usage than controls (37% vs. 3%) [37]. This is consistent with Patel et al., who found increased risk of acute respiratory infections in association with kerosene and other oils as a lighting source compared to electric and solar lighting, after adjusting for household environment, pollutants, place of cooking and socio-economic and demographic characteristics (OR 1.07, 95% CI 1.05–1.10) [27]. However, participants using any other source of light apart from kerosene and other oils had a reduced risk of acute respiratory infections compared to those using electric and solar lighting (OR = 0.91, 95% CI 0.84–0.99) [27].

One case-control study reported that childhood burns were associated with kerosene lamp usage (OR = 3.16, 95% CI 1.58–6.35) and 68% of burns in the sample could be prevented if this light source was completely eliminated from the home [38]. Similarly, Chen et al. found that compared to those using low quality light sources, mainly kerosene candles, participants adopting the use of a home solar lighting system reported decreased risk of burns by a light source (by 6.5 percentage points) and a cough (by 9.3 percentage points) [33].

Three studies explored injuries in relation to lighting amongst elderly populations. One study reported poor household lighting as a risk for home injuries, mainly falls and burns (OR = 3.00, 95% CI 1.41–6.38) [39]. The other two studies explored the association of lighting of stairs with falls. Shi et al. reported protective effects of sufficient lighting (OR 0.45, 95% CI 0.21–0.96) [40], and Isberner et al. found a non-significant increased risk with poor lighting (OR 3.31, 95% CI 0.63–17.36) [41].

A cross-sectional study conducted amongst Polish children aged six to eighteen years compared the effect of different types of electric lights in different rooms in their homes on the prevalence of refractive errors [42]. Refractive errors are caused by a defect in the eye shape which results in difficulty focusing on an image, causing blurred vision [55]. Only a higher prevalence of hyperopia (farsightedness) amongst those using fluorescent lights compared to incandescent lights (15% vs. 11%) in the kitchen were statistically significant of the five refractive errors assessed in the study [42].

##### Mental Health

A cross-over study explored the effect of colour temperature changes in the communal areas of seven care homes. The greater the colour temperature the cooler the light. The study found that residents exposed to a cooler, blue enriched 17,000 K light compared with a warmer 4000 K light reported lower daytime anxiety levels, but no significant effect on other mood indicators including depression [43].

##### Sleep Health

The differential effect of the main artificial light sources including light generated by light bulbs, fluorescent light bulbs and light emitting diodes (LED) on subjective sleep quality was the subject of one study. An increased risk of worsened sleep quality was associated with the use of light bulbs (OR 3.70, 95% CI 1.1–12.6) and fluorescent light bulbs (OR 2.1, 95% CI 0.8–5.7), although the latter association was non-significant [44]. Light colour temperature has been identified as another relevant factor impacting sleep quality. Cooler light colour temperature (17,000 K light compared with 4000 K light) was associated with worsened subjective sleep quality in addition to other measures of worsened sleep (sleep efficiency, sleep time and sleep percentage), sleep latency was also worsened, however, this finding was not significant [43].

#### 3.3.3. Light at Night

Health outcomes were explored in association with lighting habits at bedtime in two studies, light intensity in nine studies and length of exposure to light in one study. Eight of the eleven (73%) studies were from the Housing Environments and Health Investigation among Japanese Older People in Nara, Kansai Region (HEIJO-KYO) cohort. There is strong evidence suggesting exposure to light at night has a negative impact on health. Out of the eleven studies exploring relationships with various health outcomes, only one study found no effect with at least one health outcome.

##### Physical Health

Two studies explored the relationship between turning on the lights during sleep time and health. Czepita et al. reported no association between students sleeping with the lights on or off and the prevalence of refractive errors [42]. A case-control study of women under seventy-five years of age found that turning on the lights more frequently during sleep time increased risk of breast cancer (OR 1.65, 95% CI 1.02–2.69) [45].

One study from the HEIJO-KYO cohort by Obayashi et al. divided night-time light levels into quartiles and reported a positive association between light at night and carotid atherosclerosis (p_trend_ = 0.002) [46]. Two further studies from the same cohort set a threshold for bedroom light at night intensity. Light at night exposure ≥3 lux was associated with a higher risk of dyslipidemia (OR 1.72, 95% CI 1.11–2.68), body mass index (OR 1.89, 95% CI 1.02–2.57) and abdominal obesity (OR 1.62, 95% CI 1.02–2.57)) [47], and light at night exposure ≥5 lux with higher night-time blood pressure (mean of participants’ measurement, adjusted for confounders-systolic: 120.8 vs. 116.5 mmHg; diastolic: 70.1 vs. 67.1 mmHg) [48]. Additionally, a significant association between evening light exposure (4-h period before bedtime) and diabetes (OR 1.72, 95% CI 1.12–2.64) was reported in a further study from the HEIJO-KYO cohort [49]. A cross-over study compared the effect of a light (1000 lux) and dark (0 lux) bedroom environment on heart function in young healthy adults. The presence of light during sleep was associated with greater sympathetic dominance, indicated by an increased low-frequency power divided by high-frequency power of heart rate variability [50].

##### Mental Health

A cross-sectional study in the HEIJO-KYO cohort reported a positive association between light at night ≥5 lux and depression (OR 1.89, 95% CI 1.10–3.25) compared to light at night <5 lux [51]. Results were consistent with a follow-up longitudinal study (hazard ratio 1.72, 95% CI 1.07–2.96) in the same cohort using the same LAN cut off measurements [52]. Longer duration (≥30 min) in high intensity light (≥10 lux) during the in-bed period was associated with increased risk of depression in the longitudinal study (OR 1.71, 95% CI 1.01–2.89) [51]. 

##### Sleep Health

A study amongst Japanese elders ≥60 years of age measured light in lux using a light meter placed near the head of the bed divided night-time light levels into quartiles. The study found a greater risk for poor sleep quality for individuals in the highest quartile compared with the lowest (OR 1.61, 95% CI 1.05–2.45, *p* < 0.001) [53]. An experimental study by Yamauchi et al. showed sleep latency and sleep efficacy were unaffected by two different light environments in the bedroom. The study compared a light environment, defined as 1000 lux of fluorescent light, with a dark environment (approximately 0 lux) measured using a light meter [50]. The light environment was, however, associated with an increased apnea-hypopnea index [50]. The relationship between evening light exposure was also investigated in association with sleep onset latency by a longitudinal study. The study reported evidence of a significant positive relationship, exposure to greater intensity light during the evening was associated with sleep onset latency (regression co-efficient 0.133, 95% CI 0.020–0.247) [54].

## 4. Discussion

### 4.1. Impact of Lighting in the Home on Health

This systematic review synthesised the existing evidence on links between lighting in the home and health. Though limited in number, the available studies evaluated a range of lighting types (natural light, artificial light and light at night) across twenty-two specific health outcomes. Of the twenty-eight studies included in this review, twenty-five studies observed an association of lighting exposure on at least one health outcome; five of these studies investigated natural light, ten artificial light and ten light at night.

#### 4.1.1. Natural Light

In general, the included studies showed positive associations of natural light exposure and improved health across all health domains (physical, mental and sleep health). Adequate natural light at home has been found to be protective for various health outcomes including tuberculosis, leprosy, depression, mood, falls and sleep. These findings are in line with previous studies conducted in settings other than the home, including offices and hospitals. For instance, in offices, evidence suggests workers with less sunlight exposure have worse self-reported sleep quality [56] and mood [57]. Three systematic reviews focusing on hospital settings identified positive effects on depression in patients with diagnosed depressive illnesses attributable to increased sunlight exposure [58,59,60]. Findings also suggest that exposure to sunlight can improve sleep amongst all hospital in-patients [58,60].

Our systematic review also identified protective effects of natural light with respect to infectious diseases, possibly due to sunlight’s ability to kill bacteria [61]. Ultraviolet light might act as a natural disinfectant, by weakening and damaging bacteria, causing mutations that limit their ability to reproduce and survive [62]. This disinfectant effect has been found to persists via indirect sunlight exposure through glass [18,63] and windows in homes [64].

#### 4.1.2. Artificial Light

Studies included under the artificial light category used a diverse range of methods to measure exposure to artificial lighting. Methods varied from different sources of artificial lighting (e.g., fuel based, electric, and solar), different electrical lighting equipment (e.g., light bulbs, LED), the adequacy of the lighting available and adjustments to existing electrical lighting (e.g., provision of additional lamps to the living room), all of which showed an impact on health. The majority of studies focused on differential effects on health due to different sources of artificial lighting in the home. There is clear evidence that use of fuel based light sources negatively impact health. In the developing world, 860 million people lack access to electricity [65], as such fuel-based lighting is the common method to illuminate the home. Our review included five studies which evaluated fuel-based lighting compared to electric/solar lighting which were carried out in LMICs. Of those, four showed that individuals using fuel-based lighting compared to electric or solar lighting are more likely to suffer from respiratory diseases and burns. This is consistent with a comprehensive review by Mills et al. on the health impacts of fuel-based lighting [66,67]. Studies have shown that fuel-based indoor lighting choices significantly contribute to the level of indoor air pollution [68]. Fuel based lighting releases particulate matter, volatile organic compounds and other harmful pollutants when burned, and inhalation of these particles into the lungs results in respiratory disease such as acute respiratory lung infections and lung cancer [69,70,71], especially in homes with poor ventilation.

Burns are one of the top causes of non-fatal injury in children [72]. The use of fuel-based lighting sources can result in burn injuries, for example via overturned kerosene lamps, with a significant proportion of burns occurring amongst children. The placement of lamps is, therefore, an important consideration.

The type of electric lighting (bulb type and colour temperature) in the home can also impact health. One of the studies on light bulb types included in this review [44] reported worse sleep quality for a light bulb versus LED lighting. It is worth noting, this study did not classify what specific light-bulb types were considered under “light bulb”. However, in Japan, where the study took place, this term is often used to reflect incandescent light bulb [73].

Only one study reported on colour temperature. Worsened sleep quality but lower daytime anxiety levels were reported when exposed to cooler lights compared to warmer light during the 12-week cross-over study [43]. Light naturally contains a spectrum of colours. The light falling on the eye has an important role in regulating the circadian rhythm. Melanopsin, a photoreceptor in the eye, responds to rich blue light and signals the suppression of melatonin [74]. Sunlight has lower wavelengths during the day corresponding with a bluer light [75]. This exposure to daylight helps to stay alert, while evening exposure to light bulbs containing high levels of blue light signals processes affecting melatonin release and negatively impact sleep [9,76,77]. However, in the study only overhead lights in communal areas (lounge and dining room) were adjusted, and although only residents that frequented these areas were eligible, participants were not be exposed to the intervention in the evening upon returning to their bedroom [43].

#### 4.1.3. Light at Night

Effects of light at night were generally consistently associated with the analysed health outcomes. The majority of the evidence came from the HEIJO-KYO cohort. Results consistently showed high levels of indoor light at night was associated with negative health outcomes (including sleep and metabolic disorders such as obesity, diabetes and dyslipidaemia). The studies conducted on the HEIJO-KYO cohort were of good quality with rigorous methodology, including objective measures of evening and bedroom night-time light intensity with a light meter. Although these results are restricted to a sub-population of home-dwelling Japanese elders, they are in line with the findings of another review [24]. Cho et al. reviewed the effects of artificial light at night across the general population and identified that chronic light at night exposure could negatively impact sleep and other physiological functions [24]. Their review, however, incorporated studies using satellite imagery to measure the outdoor night-time light level. Studies using this measurement method were not eligible for inclusion in our systematic review as it lacks consideration for individual level factors, like window covering practices with blinds and location of the bedroom in relation to streetlights, and as such is not always reliable to represent an individual’s exposure to light within the home [78,79]. Nonetheless, there is still further evidence available supporting the negative impact of light at night within the home on health and sleep in particular [22,76,80,81]. A clear biological explanation for this association exists. Bright light during the night is ill-timed, causing disruptive effects on the circadian rhythm, through suppression of melatonin and subsequently affecting sleep and other metabolic processes [82]. Although this systematic review, sought to evaluate the effect of lighting in the home on melatonin itself in addition to all health outcomes, no studies evaluating the effects of lighting in the home with melatonin were identified or eligible.

### 4.2. Strengths and Limitations

Our systematic review examines lighting within the home and associated health outcomes in residents. Included studies analysed different types of light exposure which are conceptually different and use different exposure metrics to quantify light exposure. Direct comparison between studies and meta-analysis was consequently not feasible. Instead, we synthesised the strength of evidence of association for different types of light within broad health domains, which enabled us to establish general links with these domains but not with specific health outcomes.

Multiple databases were comprehensively and systematically searched for existing literature on lighting in the home and health. Included studies were restricted to the English language and we did not seek grey literature which may have led to the exclusion of some relevant studies. Publication bias was not formally assessed due to the limited number of studies overall and the limited number of studies within each category of lighting and health.

None of the included studies were of poor quality, but methodological limitations were present. Many of the studies used self-reported measurements of health outcomes which could introduce recall bias. A number of studies did not take physical measurements of light, instead relying on subjective reports from participants which may be under or over-estimating exposure. Furthermore, in five studies [31,32,39,40,41] the type of lighting studied was not clear or unclassified. In these cases, categorisation was made according to the authors’ judgement based on the timing of the light exposure and the location of the exposure within the house.

### 4.3. Future Research

A gap highlighted by this review is that lighting in the home and specific health outcomes have not been well studied despite strong hypotheses for such links. There appears to be more breadth than depth when investigating the relationship of indoor lighting with health (Table 2). In many cases, specific health outcomes were only evaluated by one study. These specific health outcomes should be further explored to ensure the consistency of findings. In addition, there should be further research of health outcome associated with light at night measured in the home amongst other populations to identify whether the results for the HEIJO-KYO cohort are generalisable to other socioeconomic and cultural settings.

### 4.4. Wider Policy Implications

Policy decisions surrounding lighting in the home are generally viewed from a cost-saving perspective. The focus is often on energy efficiency [83], mostly by means of bulb choice; but also via improving daylight efficiency to reduce the use of artificial lighting, which is often powered by electricity. Selection of lights that appeal aesthetically and the functionality of lighting (for example, general ambient lighting, task lighting directed to enable the completion of a goal and accent lighting to highlight a certain area) are other aspects frequently considered by building and lighting designers [84]. The evidence emerging from our review suggests stronger emphasis should be placed on the physiological impacts of lighting in homes. Some housing quality guidance tools (e.g., the National Healthy Housing Standard [85] and Housing Health and Safety Rating System [86]) consider the role of light on health in their assessments of housing but these often concern only the commonly referenced impacts on injuries and depression. The current evidence base is not substantial enough to support recommendations for policy and guidance to improve specific health outcomes that emerge as being associated with lighting in the home. It is clear, however, that there is an intersection between lighting and health and, as such, consideration for health should be made when constructing and designing homes.

In the meantime, based on the findings from this review simple measures around the home can be taken by residents to ensure optimal lighting conditions for their health. Actions such as keeping curtains open during the day to allow natural light into the home, making improvements to poor lighting (particularly around the stairs) and sleeping in a room of darkness or with an eye mask are encouraged. In addition, safe and healthier alternatives to fuel-based lighting such as off-grid electrical systems (e.g., solar power) should be considered in settings without electricity access. In cases where this is not feasible, e.g., due to initial set up costs, consideration should be given to the placement of fuel-based lighting sources in the home to prevent accidental injuries.

## 5. Conclusions

This review found that some types of lighting in the home can negatively impact health but identified only a limited number of studies at present that explore this relationship in different domains of light and health. Our findings warrant further attention for research as evidence on lighting in the home and its association with specific health outcomes is required to better inform housing quality assessments, lighting practises in the home and housing policies to ensure the home is a safe and healthy space.

## Figures and Tables

**Figure 1 ijerph-18-00609-f001:**
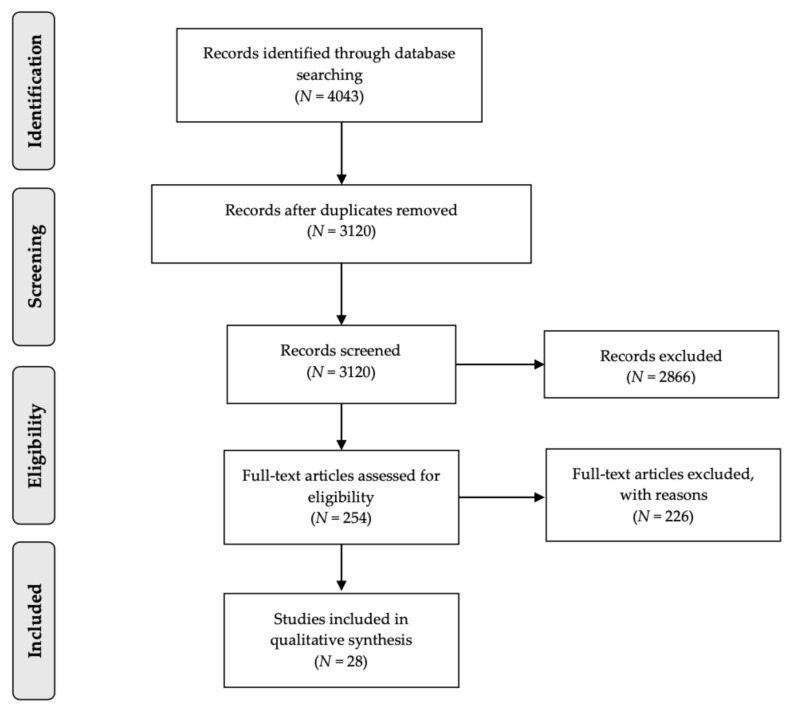
Preferred Reporting Items for Systematic Reviews and Meta Analyses flow diagram.

**Table 1 ijerph-18-00609-t001:** Key characteristics of studies included in the systematic review.

First Author, Year Reference	Study Design	Country	Sample Size	Age (Years)	Lighting Exposure	Health Domain	Health Outcome	Main Finding	Quality Score
**NATURAL LIGHT**									
Rahayu, 2015 [28]	Case-control	Indonesia	212	Adults	Subjective: Presence of sunlight	PH	OM: Tuberculosis	↑ Presence of sunlight in the house protective against tuberculosis (OR 0.06, 95% CI 0.00–0.67) *	Fair
Kumar, 2001 [29]	Cross-sectional	India	13,320	All	Subjective: Insufficient household light exposure	PH	OM: Leprosy	↑ Persons living in houses with insufficient sunlight exposure observed to be more afflicted by leprosy (OR 1.57, 95% CI 0.84–2.88)	Fair
Brown, 2011 [30]	Cross-sectional	Lithuania, Switzerland, Italy, Germany, Portugal, Hungary, Slovakia and France	6017	≥18	Subjective: Inadequate residential light	PH MH	SR: Falls and depression	↑ Participants reporting inadequate natural light in dwelling more likely to report falls (OR 1.5, 95% CI 1.2–1.9) * and depression (OR 1.4, 95% CI 1.2–1.7) *	Fair
Ichimori, 2013 [31]	Cross-sectional	Japan	24	76–90	Objective: Daytime illuminance	PH MH	SR: Physical health and depression	- No relationship between illuminance and physical health ↑ Time exposed to light over 400 lx and depression scores *	Fair
Youngstedt, 2004 [32]	Cross-sectional	USA	459	50–81	Objective: Morning illuminance	PH SH	SR: Mood SR and OM: Sleep	↑ Morning illumination moderately associated with improved mood * and sleep	Fair
**ARTIFICIAL LIGHT**									
Chen, 2017 [33]	Case-control	Uganda	934	NR	Treatment: Solar home lighting system Comparison: Low quality sources	PH GH	SR: Burns, cough and personal health	↑ Burns by lighting source 6.5 p.p. less; cough 9.3 p.p. less; and self-reported health 35.2 p.p. higher among households with solar home lighting system	Fair
Brunnstrom, 2004 [34]	Randomised trial	Sweden	46	20–90	Intervention: Living room adjustment-50 Watts halogen, 12 Volt standard floor lamp	GH	SR: General health and depressed mood	↑ Improvement in general health *p* < 0.01 and depressed mood *p* < 0.04 after the adaptation was found for the intervention group	Fair
**First Author, Year**	**Study Design**	**Country**	**Sample Size**	**Age (Years)**	**Lighting Exposure**	**Health Domain**	**Health Outcome**	**Main Finding**	**Quality Score**
Falkenberg, 2019 [35]	Randomised trial	Norway	60	77	Intervention: Providing lamps to achieve recommended living room lighting levels (200 lux)	GH	SR: Visual health and general health	- Self-reported visual problems and health unchanged in both groups during the intervention	Good
Woldesemayat, 2014 [36]	Case-control	Ethiopia	1154	Adult	Kerosene lamps, electricity, others	PH	OM: Pulmonary tuberculosis	- Kerosene lamps used for lighting by 73% cases and 71.5% controls, electric lighting used by 24.5% cases and 26.6% controls. The remaining participants used other kerosene-based or other light sources	Fair
Savitha, 2007 [37]	Case-control	India	208	0–5	Kerosene lamps, electricity	PH	OM: Acute lower respiratory infection (ALRI)	↑ 36.54% of ALRI cases used kerosene lamps for lighting compared to 2.88% of controls, which used electric lighting	Fair
Patel, 2019 [27]	Cross-sectional	India	932,341	0–59 months	Electricity and solar, kerosene and other oils, others	PH	SR: Acute respiratory infection (ARI)	↑ Kerosene and other sources for lighting have higher (OR 1.07, 95% CI 1.05–1.10) * for ARI compared to electric and solar lighting	Fair
Mashreky, 2010 [38]	Case–control	Bangladesh	840	<10	Use of traditional kerosene lamp (kupi bati)	PH	OM: Burn	↑ Using a kupi bati increased risk of burn (OR 3.16, 95% CI 1.58–6.35) * with attributable risk of 68.38%	Fair
Camilloni, 2011 [39]	Case-control	Italy	74	65–85	Subjective: Poor lighting	PH	SR: Home injuries	↑ Poor household illumination associated with home injuries (OR 3.00, 95% CI 1.41–6.38) *	Fair
Shi, 2014 [40]	Cross-sectional	China	472	Adults	Subjective: Sufficient lighting for stairway	PH	SR: Falls	↑ Sufficient lighting for stairway can lower the risk for a single fall (OR 0.45, 95% CI 0.21–0.96) *	Good
Isberner, 1998 [41]	Case-control	USA	90	≥60	Subjective: Poor lighting at stairs	PH	SR: Falls	↑ Participants with poor lighting at stairs had a higher chance of falling (OR 3.31, 95% CI 0.63–17.36)	Fair
**First Author, Year**	**Study Design**	**Country**	**Sample Size**	**Age (Years)**	**Lighting Exposure**	**Health Domain**	**Health Outcome**	**Main Finding**	**Quality Score**
Czepita, 2004 [42]	Cross-sectional	Poland	3636	6–18	Type of lighting: Fluorescent or incandescent in living room, dining room, child’s room, parent’s room, kitchen and bathroom.	PH	OM: Refractive error: emmetropia, myopia, hyperopia, astigmatism and anisometropia	↑ Higher prevalence of hyperopia with fluorescent lamp in kitchen (*p* < 0.01) * - No statistically significant findings for other exposure-outcome combinations	Fair
Hopkins, 2017 [43]	Crossover	UK	80	>60	Blue-enriched white lighting (17,000 K ≃ 900 lux), white lighting (4000 K ≃200 lux)	MH SH	SR: Mood SR and OM: Sleep	↑ Blue-enriched lighting reduced anxiety, sleep efficiency and quality * ↑ Blue-enriched light increased night-time activity *	Fair
Kayaba, 2014 [44]	Cross-sectional	Japan	351	20–70	Light-emitting diode (LED), light bulb, fluorescent light	SH	SR: Sleep quality	Compared with LED lighting: ↑ Light bulbs (OR: 3.7, 95% CI 1.1–12.6) * were risk factors for variable sleep quality - Fluorescent lighting produced no significant results (OR 2.1, 95% CI 0.8–5.7)	Fair
**LIGHT AT NIGHT**									
Czepita, 2004 [42]	Cross-sectional	Poland	3636	6–18	Lighting habit: Sleeping in darkness or with the light on	PH	OM: Refractive error (emmetropia, myopia, hyperopia, astigmatism and anisometropia)	- No relationship between prevalence of refractive error and sleeping with the light turned on or off at night	Fair
O’Leary, 2006 [45]	Case-control	USA	1161	<75	Lighting habit during sleep hours	PH	OM: Breast cancer	↑ Increased risk of breast cancer for women who frequently turned on lights at home during sleep hours (OR 1.65, 95% CI 1.02–2.69) *	Fair
**First Author, Year**	**Study Design**	**Country**	**Sample Size**	**Age (Years)**	**Lighting Exposure**	**Health Domain**	**Health Outcome**	**Main Finding**	**Quality Score**
Obayashi, 2015 [46]	Cross-sectional	Japan	700	≥60	Objective: Indoor illumination level	PH	OM: Carotid atherosclerosis	↑ With each quartile increase in light exposure, mean carotid intima-media thickness increased (p_trend_ = 0.002) *	Good
Obayashi, 2013 [47]	Cross-sectional	Japan	528	≥60	Objective: Indoor illumination level	PH	SR and OM: Obesity and dyslipidaemia	↑ Light intensity and Body Mass Index (OR 1.89, 95% CI 1.02–2.57) *; abdominal obesity (OR 1.62, 95% CI 1.02–2.57) *; and dyslipidaemia (OR 1.72, 95% CI 1.11–2.68) *	Good
Obayashi, 2014 [48]	Cross-sectional	Japan	528	≥60	Objective: Indoor illumination level	PH	OM: Night-time blood pressure	↑ Light intensity (≥5 lux) higher night-time systolic BP (adjusted mean: 120.8 vs. 116.5 mmHg) and diastolic BP (70.1 vs. 67.1 mmHg) compared with group <5lux	Good
Obayashi, 2014 [49]	Cross-sectional	Japan	513	≥60	Objective: Indoor illumination level	PH	OM: Diabetes	↑ Brighter evening light amounts and increase in diabetes prevalence (OR 1.72, 95% CI 1.12–2.64) *	Good
Yamauchi, 2014 [50]	Crossover	Japan	17	Adult	Light environment: 1000 lux, Dark: 0 lux	PH SH	SR and OM: Sleep (efficacy, latency and apnea) and heart rate variability	↑ Higher low-frequency power divided by high-frequency ratio power in the analysis of heart rate variability and apnea-hypopnea index in the light environment * - No other differences in sleep in the different light environments	Fair
Obayashi, 2013 [51]	Cross-sectional	Japan	516	≥60	Objective: Indoor illumination level	MH	SR: Depression	↑ Higher prevalence of light intensity ≥5lux in the depressed group compared with that in the nondepressed group (OR 1.89, 95% CI 1.10–3.25) * ↑ Light at night 10 lux ≥30 min is a risk for depressive symptoms (OR 1.71, 95% CI 1.01–2.89) *	Good
Obayashi, 2018 [52]	Longitudinal	Japan	863	≥60	Objective: Indoor illumination level	MH	SR: Depression	↑ Light intensity (≥5 Lux) and higher depression risk (HR 1.78, 95% CI 1.07–2.96) *	Good
Obayashi, 2014 [53]	Cross-sectional	Japan	857	≥60	Objective: Indoor illumination level	SH	SR and OM: Sleep quality	↑ Highest quartile of light intensity showed higher odds for insomnia (OR 1.61, 95% CI 1.05–2.45) *; higher OR for insomnia with each quartile increase in light exposure (p_trend_ = 0.043) *	Good
Obayashi, 2014 [54]	Longitudinal	Japan	192	≥60	Objective: Indoor illumination level	SH	OM: Sleep onset latency	↑ Brighter evening light amounts and longer sleep onset latency (regression co-efficient 0.133, 95% CI 0.020–0.247) *	Good

Abbreviations: PH = physical health, MH = mental health, SH = sleep health, GH = general health, OM = objectively measured, SR = self-reported, NR = not reported, OR = odds ratio, HR = hazard ratio, p.p = percentage points, **↑** = expected direction of association, **↓ =** unexpected direction of association, - **=** no association * = statistically significant with *p* < 0.05.

**Table 2 ijerph-18-00609-t002:** Specific health outcomes in the broad health domains.

Physical Health	Mental Health	Sleep Health
Injury (including falls and burns) Leprosy Tuberculosis Acute respiratory infections Cough Breast cancer Visual health (including refractive errors) Atherosclerosis Obesity Dyslipidaemia Diabetes Blood pressure Heart function	Depression Anxiety Mood	Sleep quality Sleep latency Sleep efficacy Sleep time Sleep percentage Sleep apnoea

## Data Availability

No new data were created or analysed in this study. Data sharing is not applicable to this article.

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
