# Peer review of "Lighting in the Home and Health: A Systematic Review"

_ijerph, 2021, doi:10.3390/ijerph18020609_

Round 1
Reviewer 1 Report
This article deal with the relationship between lighting in the home and health, and recommends areas for future research.
Three databases were searched for relevant literature using pre-defined inclusion criteria. The study quality was assessed using the Newcastle Ottawa Scale. Extracted data were qualitatively synthesised according to type of lighting (natural light, artificial light and light at night) and stratified by broad health domains (physical, mental and sleep health).
Some types of lighting in the home can negatively affect health but the current evidence base is limited to a small number of studies in different domains of light and health.
Further research surrounding specific health outcomes is required to better inform housing quality assessments and lighting practises in the home.
The limitations imposed must be made clear, they can provide interesting but transversal information: lighting and health in architecture (buildings).
The study has potential to be a useful contribution to the journal of Int. J. Environ. Res. Public Health. The topic of lighting and health is meaningful and relevant for an international audience.
Pg 3 line 99 Define BS
Pg 3 line 102 Define DF
Pg 4 line 141 capital initials: “low-middle income countries”
Pg 10 line 33 capital initials: Odds Ratio (OR)
There are prior studies related to natural lighting (daylight) not included for example: “Daylight and health: A review of the evidence and consequences for the built environment” may be interesting to include it to conclude. Maybe building or built is a key word missing in the search.
Pg 15 line 290 Conclusions
Clarify “This review found that lighting in the home can negatively impact health,” for example “some types of lighting”
The conclusions are very general, specify research parameters and adequate lighting policies based on the results (lighting and health domains), recommends areas for future research according with abstract and Limitations of the study.
Reviewer 2 Report
The authors have demonstrated a good rationale for the study. The aim is very broad but all the same the study is well conducted and manuscript well written. Therefore, I do not have many comments. The paper should open up more related research in the coming years and hence will be well cited..
The introduction is bit too long. Light, which is the exposure of interest is first mentioned in paragraph 4. While the other factors define adequate housing, they are not the focus of this research. The authors should contract those first three paragraphs into a small one.
Does the Rahayu et al. study really show statistical significance, even if they reported so?
Reviewer 3 Report
The paper is interesting and the topic of lighting in the home is an actual argument of debate.
Methodology utilized is good and relevant studies are present, adequately categorized and summarized.
The paper is well organized and written making reading stimulated and pleasant.
I suggest to insert in the tables one or more columns relating to the specification of the methodology used in each study (e.g . subjective reports or not) for the detection of health outcomes and the measurement of light.
Reviewer 4 Report
Major revision required
The definition of light has not been clarified with enough detail. Is it the absence of light or is it the fuel used for lightening - gas, electricity, sun etc? The authors do not seem to have considered that different concepts that are operating in this review.
The study has not been carefully conceptualised. For example included studies are those describing the fuel type used for lighting (e.g Patel etal), compared to studies that described improvements in illumination to reduce falls (Brown etal). These are very different exposures. It does not make sense epidemiologically to include such studies together in a single review. One is the comparison of exposure to fuel (pertaining to household air pollution) while the other is an improvement in the environment (illumination). Conceptually these are distinct. Further, it also seems odd to include studies that examined night-time light exposure on blood pressure to exposure to sunlight to reduce leprosy, or type of fuel for lighting to reduce burns. These cannot be compared.
The introduction is too long and does not provide a thorough review of the background literature and or the biological plausibility of the effect of light on health outcomes. The historical paragraph is superfluous. The background needs to be shortened and reorientated to focus on light as an exposure, but should be limited to light (and not fuel type in relation to household air pollution).
The search strategy is inadequately described, and should be included in the main paper. A high-quality description of the methods is critical for readers to determine the quality of the review. This needs careful and thorough revision.
It seems that there has not been careful consideration of the inclusion criteria. Eg. The inclusion of the studies that although include lighting are conceptually different exposures.
Round 2
Reviewer 4 Report
You can now accept this manuscript, the authors have satisfactorily addressed my comments.